# The Potential of Single-Transcription Factor Gene Expression by RT-qPCR for Subtyping Small Cell Lung Cancer

**DOI:** 10.3390/ijms26031293

**Published:** 2025-02-03

**Authors:** Albert Iñañez, Raúl del Rey-Vergara, Fabricio Quimis, Pedro Rocha, Miguel Galindo, Sílvia Menéndez, Laura Masfarré, Ignacio Sánchez, Marina Carpes, Carlos Martínez, Sandra Pérez-Buira, Federico Rojo, Ana Rovira, Edurne Arriola

**Affiliations:** 1Cancer Research Program, Hospital del Mar Research Institute, 08003 Barcelona, Spain; albert.i.blas1999@gmail.com (A.I.);; 2Department of Medical Oncology, Hospital del Mar, 08003 Barcelona, Spain; 3Department of Pathology, Hospital del Mar, 08003 Barcelona, Spain; 4Instituto Murciano de Investigación Biosanitaria IMIB-Pascual Parrilla, Pathology Core, 30120 Murcia, Spain; 5Department of Anatomy and Comparative Pathology, Facultad de Veterinaria, Universidad de Murcia, 30100 Murcia, Spain; 6Department of Pathology, IIS-Fundación Jiménez Díaz-CIBERONC, 28040 Madrid, Spain; 7Centro de Investigación Biomédica en Red de Cáncer (CIBERONC), 28029 Madrid, Spain

**Keywords:** SCLC subtypes, RT-qPCR, single-gene marker, *ASCL1*, *NEUROD1*, *POU2F3*

## Abstract

Complex RNA-seq signatures involving the transcription factors *ASCL1*, *NEUROD1*, and *POU2F3* classify Small Cell Lung Cancer (SCLC) into four subtypes: SCLC-A, SCLC-N, SCLC-P, and SCLC-I (triple negative or inflamed). Preliminary studies suggest that identifying these subtypes can guide targeted therapies and potentially improve outcomes. This study aims to evaluate whether the expression levels of these three key transcription factors can effectively classify SCLC subtypes, comparable to the use of individual antibodies in immunohistochemical (IHC) analysis of formalin-fixed, paraffin-embedded (FFPE) tumor samples. We analyzed preclinical models of increasing complexity, including eleven human and five mouse SCLC cell lines, six patient-derived xenografts (PDXs), and two circulating tumor cell (CTC)-derived xenografts (CDXs) generated in our laboratory. RT-qPCR conditions were established to detect the expression levels of *ASCL1*, *NEUROD1*, and *POU2F3*. Additionally, protein-level analysis was performed using Western blot for cell lines and IHC for FFPE samples of PDX and CDX tumors, following our experience with patient tumor samples from the CANTABRICO trial (NCT04712903). We found that the analyzed SCLC cell line models predominantly expressed *ASCL1*, *NEUROD1*, and *POU2F3*, or showed no expression, as identified by RT-qPCR, consistently matching the previously assigned subtypes for each cell line. The classification of PDX and CDX models demonstrated consistency between RT-qPCR and IHC analyses of the transcription factors. Our results show that single-gene analysis by RT-qPCR from FFPE-extracted RNA simplifies SCLC subtype classification. This approach provides a cost-effective alternative to IHC staining or expensive multi-gene RNA sequencing panels, making SCLC subtyping more accessible for both preclinical research and clinical applications.

## 1. Introduction

Small Cell Lung Cancer (SCLC) is a highly aggressive and metastatic malignancy characterized by a poor prognosis and high mortality rate [1]. Diagnosis primarily relies on histological evaluation using hematoxylin and eosin staining, complemented with the application of neuroendocrine immunohistochemistry (IHC) markers such as synaptophysin, chromogranin A, CD56, and the proliferative marker Ki67 [2].

Despite decades of research, advances in SCLC treatment remain limited. The addition of anti-PD-L1 immunotherapy to chemotherapy extends median overall survival by 2 months (10 to 12 months) [3,4] with 5-year survival rates below 7% [5]. Moreover, no validated predictive biomarkers exist to guide anti-PD-L1 therapy in SCLC [6,7]. This limited progress highlights the need for personalized treatment strategies based on tumor molecular characteristics.

SCLC is challenging to classify by driver mutations due to its genomic instability and a lack of actionable targets. Unlike cancers with treatable drivers (e.g., EGFR or ALK in non-small cell lung cancer), SCLC often harbors mutations in tumor suppressors such as TP53 and RB1, which remain untargetable [8]. The molecular subclassification of SCLC has advanced significantly since 1985 when Gazdar et al. [9] first divided SCLC into “classic” and “variant” subtypes. This classification was based on an in-depth analysis of 50 SCLC cell lines, identifying unique biochemical markers, morphological features, and growth properties.

Recent efforts focus on transcriptional profiling for identifying therapeutic targets and subtypes [10] with studies defining SCLC subtypes by differential expression of transcriptional factors, including achaete-scute homolog 1 (*ASCL1*), neurogenic differentiation factor 1 (*NEUROD1*) [11], POU class 2 homeobox 3 (*POU2F3*) [12], and yes-associated protein 1 (YAP1) [13]. There is ongoing debate regarding two molecular classifications of SCLC, which share three subtypes (*ASCL1*, *NEUROD1*, and *POU2F3*) but differ in the fourth. Rudin et al. [14] defined four subtypes based on the relative expression of four transcriptional factors: SCLC-A (*ASCL1*), SCLC-N (*NEUROD1*), SCLC-P (*POU2F3*), and SCLC-Y (YAP1). In contrast, Gay et al. [10] proposed four groups based on the relative expression of three transcriptional factors: SCLC-A (*ASCL1*), SCLC-N (*NEUROD1*), SCLC-P (*POU2F3*), and SCLC-I. SCLC-I, referred to as “triple-negative” or “inflamed”, lacks expression of *ASCL1*, *NEUROD1*, and *POU2F3* and is characterized by an inflamed gene signature. Gay et al.’s [10] classification is now more widely accepted, primarily due to challenges in consistently defining the SCLC-Y subtype [15,16]. However, some controversy in this field remains.

These subtypes can also be classified based on lineage-specific markers such as neuroendocrine (NE) or non-neuroendocrine (non-NE). NE subtypes, SCLC-A and SCLC-N, are defined by *ASCL1* and NEUROD1, which drive NE differentiation at different stages [17]. Although SCLC-A and SCLC-N are distinct subtypes, some tumors exhibit a double-positive phenotype (SCLC-AN) [16]. Non-NE subtypes, including SCLC-P and either SCLC-Y or SCLC-I, lack high expression of NE markers.

These subtypes have the potential to transform SCLC research, emphasizing the need to align therapies with molecular subtypes. Accurate subtype classification is crucial for understanding tumor heterogeneity and tailoring therapeutic strategies. Immunotherapy, while limited success in SCLC, may be more effective in subtypes like SCLC-I, which exhibit greater immune cell infiltration [18,19].

Recent studies suggest that targeting Delta-like protein (DLL3), a direct transcriptional target of *ASCL1*, may yield promising results, with SCLC-A patients potentially showing greater responsiveness [20]. The predictive and prognostic values of this molecular classification are under investigation, with ongoing research aiming to refine subtypes and enable more precise SCLC treatments.

Traditionally, SCLC classification has relied on either IHC or complex gene expression signatures [10,14], both of which present limitations. IHC is semi-quantitative, requires significant expertise for accurate interpretation, and is subjected to variability in staining intensity over time, influenced by batch-to-batch differences, reagent consistency, and procedural factors. Moreover, cytological samples, which are commonly encountered in SCLC, are valuable for morphological evaluation and diagnosis. However, their lack of tissue architecture, combined with the intrinsic heterogeneity and variants of SCLC, complicates the interpretation of diagnostic confirmatory IHC, particularly for cytoplasmic markers. Therefore, additional techniques to validate these results could be beneficial for this type of sample.

On the other hand, RNA sequencing, while comprehensive, is costly, time-consuming, and less effective when using degraded RNA from FFPE tissues [21], compared to higher-quality RNA obtained from fresh-frozen samples [22]. The use of these different sample types across studies further complicates reproducibility and consistency in molecular classification.

This study evaluates the feasibility of using RT-qPCR for highly quantitative, reliable, and cost-effective analysis of *ASCL1*, *NEUROD1*, and *POU2F3* expressions to simplify SCLC molecular subclassification. This approach could enhance preclinical research, clinical practice, and therapeutic personalization in SCLC.

## 2. Results

As a research group committed to advancing the understanding of SCLC, we utilize a comprehensive collection of experimental models, including cell lines, patient-derived xenografts (PDXs), circulating tumor cell-derived xenografts (CTC-derived xenografts), and syngeneic mouse models, to study tumor biology and develop effective therapeutic strategies. The recent identification of distinct molecular subtypes in SCLC has gained significant attention for its potential to predict responses to immunotherapy and other treatments [10].

To align our experimental models with the molecular subtypes of SCLC, we sought to characterize the subtype of our preclinical SCLC models by examining the expression of the three defining genes: *ASCL1*, *NEUROD1*, and *POU2F3*. Using quantitative reverse transcription polymerase chain reaction (RT-qPCR), we measured the relative expression levels of these markers. We opted to use the non-A/N/P nomenclature instead of SCLC-inflamed (SCLC-I). SCLC-I is characterized by an RNA-seq profile indicating higher immune infiltration and the absence of *ASCL1*, *NEUROD1*, and *POU2F3* expression. Since our study only confirmed the absence of expression of these transcription factors without addressing immune infiltration, we refer to these samples as SCLC-non-A/N/P.

This approach is crucial for enhancing the translational relevance of our models, deepening our understanding of the biological basis of SCLC subtypes, and facilitating the development of more effective targeted therapies.

### 2.1. Subtyping Human SCLC Cell Lines

Our human SCLC cell line panel exhibits significant heterogeneity in growth patterns and morphology (Figure 1). Five of the eleven cell lines (H69, H345, H748, H82, and H526) grow in suspension, forming large floating multicellular aggregates that can be easily dissociated by pipetting. All these lines exhibit a lymphoblast-like morphology. Notably, single H82 cells are nonviable, making this cell line particularly challenging to handle. In contrast, DMS273, H1048, H196, and H69M grow as adherent cells. DMS273 and H1048 show an epithelial-like cell morphology, while H196 and H69M exhibit a fibroblast-like morphology associated with a mesenchymal phenotype. Lastly, Shp77 and H841 exhibit mixed growth properties, with both adherent epithelial-like and suspension lymphoblast-like cells. Shp77 adherent cells are loosely attached and can detach without the need for trypsin, whereas H841 has a smaller proportion of suspension cells compared to its adherent population.

For molecular classification, we characterized 11 human SCLC cell lines using RT-qPCR and Western blot to evaluate the expression of key subtype markers *ASCL1*, *NEUROD1*, and *POU2F3*. To ensure the validity of the primers and antibodies we used Shp77 (SCLC-A), H82 (SCLC-N), and H1048 (SCLC-P) as positive controls, following the classification criteria established by Matsui et al. [23].

All cell lines analyzed exhibited distinct expression patterns of *ASCL1*, *NEUROD1,* or *POU2F3*. RT-qPCR results identified four SCLC-A cell lines (Shp77, H69, H748 and H345), two SCLC-N (H82 and DMS273), two SCLC-P (H1048 and H526) and three SCLC-non-A/N/P (H841, H196, and H69M) (Figure 2A). These findings demonstrate that RT-qPCR reliably distinguishes the four subtypes in established human cell lines, aligning with previous classifications and supporting the validity of our approach.

To validate the RT-qPCR results and assess whether mRNA levels correlate with protein expression, we performed a Western blot analysis. As shown in Figure 2B, the protein expression of each marker in each cell line aligns with the subtype classifications observed in the RT-qPCR data.

### 2.2. Subtyping Murine SCLC Cell Lines

From a translational perspective, immunocompetent syngeneic mouse models provide a valuable system for studying SCLC. These models allow for the modulation and evaluation of the tumor microenvironment in response to chemotherapy and anti-PD-L1 agents (the standard of care for SCLC) either alone or in combination with targeted therapies. In this context, we had five distinct mouse SCLC cell lines (KP1, 5B, RPP631, RP-48, and RP-116), which were derived from tumors isolated from various genetically engineered mouse models (GEMMs) [24,25,26]. Similar to most human SCLC cell lines, our panel of mouse SCLC cell lines grew in suspension, with some variations among them. KP1, RPP631, and RP-116 cells formed suspension multicellular aggregates that could be easily dispersed by gentle pipetting, whereas 5B and RP-48 cells formed dense spheroids that were more challenging to disperse, requiring more aggressive mechanical dissociation using a syringe (Figure 3).

The tumorigenicity of these mouse cell lines was determined by subcutaneous injection of cells (PBS/Matrigel, 1:1) into the right flank of B6129SF1/J (KP1 and 5B) and C57BL/6J (RPP631, RP-48, and RP-116) immunocompetent mice. KP1 and 5B tumors showed the fastest tumor growth, RP-116 tumors displayed the slowest growth, and RPP631 and RP-48 tumors had an intermediate growth rate (Figure 4).

To confirm the histopathological features of SCLC, tumors were isolated and processed at the end of the tumor growth kinetics experiment, and FFPE sections were prepared. Each tumor was stained with H&E and examined by one of the pathologists on our team.

The analysis revealed that all five mouse SCLC models exhibited histopathological features consistent with those of human SCLC. These tumors were highly cellular, with limited cytoplasm, enlarged nuclei, smooth nuclear membrane, a salt-and-pepper chromatin pattern, focal areas of necrosis, and evidence of apoptosis and mitotic figures (Figure 5A). To validate the neuroendocrine origin characteristic of SCLC, we assessed the expression of the neuroendocrine marker synaptophysin. For the IHC staining setup, lung tissue was used as a negative control, and pancreas tissue as a positive control (Figure 5B). Both the KP1 and 5B cell lines, processed as cell pellets along with their corresponding tumors, showed positive staining for synaptophysin (Figure 5C). These results confirm that these models reliably recapitulate the neuroendocrine features of SCLC tumors.

We evaluated the expression of the three key genes defining the SCLC subtypes (*Ascl1*, *Neurod1*, *Pou2f3*) using RT-qPCR. The RT-qPCR results revealed higher expression of *Ascl1* compared to *Neurod1* and *Pou2f3* (Figure 6). Thus, these mouse SCLC cell lines can be classified as the SCLC-A subtype.

### 2.3. Subtyping PDX and CTC Derived Models

After optimizing RT-qPCR conditions for the detection of *ASCL1*, *NEUROD1*, and *POU2F3* expressions in established cell lines, we applied this method to the molecular subclassification of six PDX models and two CDX models established within our group. This analysis was performed using RT-qPCR on FFPE samples.

The ability to reliably analyze FFPE samples is particularly significant, as this is the standard format for preserving patient tumor specimens in pathology laboratories. This capability not only enhances the clinical applicability of our approach but also facilitates its potential integration into routine diagnostic workflows, thereby bridging experimental findings and clinical practice. We implanted tumor samples from fourteen patients with SCLC into mice to generate PDX models and inoculated circulating tumor cells (CTCs) from thirty-four patients to establish CDX models. Tumor growth was observed in 6 out of 14 PDX cases within 1–9 months and in 2 out of 34 CDX cases within 3–12 months following sample implantation, indicating a low successful implantation rate. However, only three of the six PDXs and one of the two CDXs were considered established models for our studies, as they demonstrated successful in vivo growth after one freeze–thaw cycle.

RNA yield varies depending on the sample type. From 1–2 × 10^6^ cells in established cell lines, we consistently obtained 5–50 μg of RNA of high quality (RNA integrity (RIN) = 10). FFPE samples typically yielded 4–6 μg of RNA from 4–5 sections (14 µm each), sufficient for reliable analysis using our specialized protocols. Nevertheless, this type of sample presents challenges due to RNA degradation caused by formalin fixation, paraffin embedding, and suboptimal storage conditions, leading to fragmented and lower-quality RNA (even RIN < 2).

This degradation makes it challenging, to amplify longer target sequences using PCR, necessitating adjustments such as designing primers for shorter amplicons. Initial attempts to amplify *ASCL1* and *POU2F3* using primer designed for their characterization in RNA of cell lines failed due to the excessive length of the resulting amplicon. To address this limitation and ensure efficient target amplification, primers were optimized to generate shorter amplicons of approximately 100 base pairs (bp). Specifically, primers for *ASCL1* and *POU2F3* were re-designed to meet this size requirement. Furthermore, identifying a suitable housekeeping gene presented other challenges. After evaluating various candidates (*TUBB*, *GUSB*, *TPB*, *TOP1*, *YHWAZ*, *SDHA*, *HPRT1*), *TUBB* and *TOP1* were selected as the most reliable genes for normalization in our PDX and CDX samples, enabling accurate and consistent data analysis.

RT-qPCR analysis of PDX and CDX models revealed expression patterns consistent with the subtypes identified in cell line models. Specifically, four out of eight PDX and CDX models were classified within the SCLC-A subtype (PDX4, PDX9, PDX11, and CDX17), one out of eight as SCLC-N (PDX5), and three out of eight as SCLC-non-A/N/P (PDX1, PDX3, and CDX13). These classifications were corroborated by IHC analysis (Figure 7 and Table 1). Our findings demonstrate that RT-qPCR is a feasible alternative to IHC for reducing the complexity of subtype identification and classification, including in PDX and CDX models.

## 3. Discussion

SCLC is a highly aggressive disease with a poor prognosis, emphasizing the urgent need for novel therapeutic strategies. Progress in this area depends on the availability of models that closely mimic human SCLC. Among immunocompetent SCLC mouse models, GEMMs and syngeneic mouse models are closely linked, as syngeneic models are developed using tumor cell lines originating from GEMM tumors.

While GEMMs are valuable, their limitations, including the need for continuous breeding, long latency periods, and inter-mice tumor heterogeneity, create challenges for immuno-oncology studies. Establishing cell lines from GEEMs addresses these issues, enabling the generation of syngeneic mouse models that reduce the time required to obtain treatable mice, minimize variability within treatment groups, and improve reproducibility [27]. However, syngeneic mouse models with hybrid genetic backgrounds such as KP1 and 5B models may require costly and time-consuming mouse breeding, which limits group sizes and affects experiment timing. Additionally, minimal genetic differences [28] in hybrid mice can result in unsuccessful or highly variable tumor implantation.

The recent molecular subclassification of SCLC into four subtypes, each characterized by the predominantly expression of a specific transcription factor, SCLC-A (*ASCL1*-driven), SCLC-N (*NEUROD1*-driven), SCLC-P (*POU2F3*-driven), and SCLC-I (inflamed or “triple-negative” subtype), offers a framework for understanding its heterogeneity and guiding subtype-specific therapeutic strategies. Our study demonstrates that RT-qPCR can reliably classify SCLC subtypes based on the expression of *ASCL1*, *NEUROD1*, and *POU2F3*, offering an objective and quantitative alternative to IHC or complex RNA-seq signatures. This simplified method is particularly advantageous for use in preclinical research and for laboratories with limited access to advanced sequencing technology, high-quality antibodies, or extensive resources. We observed a high concordance between RT-qPCR results and protein expression across SCLC cell lines, as well as PDX and CDX models, further supporting RT-qPCR as a practical, consistent, and feasible tool for SCLC subtyping.

RT-qPCR combines precise quantification, high sensitivity, and reproducibility, due to the standardized nature of the technique, which minimizes variability often observed in IHC due to differences in staining protocols, antibody specificity, and subjective interpretation. This positions RT-qPCR as a simplified, cost-effective alternative for SCLC subclassification, enhancing accessibility across diverse research settings. Single-gene RT-qPCR analysis provides comparable accuracy with fewer resources and technical demands. Western blot analysis can also serve as well as a complementary tool to confirm protein expression, albeit being less quantitative. Together with RT-qPCR, these methods provide a robust and consistent approach to the molecular classification of SCLC subtypes.

A limitation is that RNA extraction from FFPE samples often yields low-quality RNA with reduced RIN values, which may impede the amplification of longer RNA fragments. Thanks to the RT-qPCR technique, this issue can be easily mitigated. In contrast, RNA-seq faces significant limitations when using FFPE, as the quality of the results is markedly inferior compared to those obtained from fresh samples but provides reliable results if the RNA is sufficiently preserved [29].

Looking ahead to personalized medicine, molecular subtyping based on transcription factor expression is likely to become a critical component of SCLC management. Given the accuracy and reproducibility of RT-qPCR, this methodology could be supplementary to IHC for molecular classification of SCLC.

Recent evidence suggests that SCLC subtypes classified by transcription factor expression may fail to capture the full extent of biological complexity within subtypes. In a recent study, Liu et al. [30] applied non-negative matrix factorization (NMF) to multi-omics datasets, identifying four distinct molecular subtypes (NMF1, NMF2, NMF3, and NMF4) based on proteogenomic data: these NMF-defined subtypes do not align directly with traditional transcription factor-based subtypes but provide more detailed insights into biological and therapeutic heterogeneity that might not be apparent from transcription factors expression alone. By integrating proteomic and transcriptional profiles, this study enriches the understanding by revealing additional layers of biological complexity, facilitating the development of more precise therapeutic strategies within each primary transcription factor-defined category.

Our RT-qPCR method for classifying SCLC subtypes represents a convenient tool in the precise characterization of experimental models, including cell lines, PDXs, and CDXs. Standardized classification is essential to ensure consistency across SCLC research. The standardization of subtype classification using our approach is particularly significant for ensuring consistency and comparability across research efforts within the SCLC field. By adopting a uniform classification system, discrepancies arising from varied methodologies employed by different laboratories can be minimized. This uniformity not only enhances the validity of inter-study comparisons but also improves the reliability of preclinical models as platforms for translational research and drug discovery. Several research groups are advancing SCLC studies by developing and utilizing diverse models such as GEMMs [31,32] as well as PDXs and CTC-derived models [31,33]. These efforts highlight the need for a standardized SCLC classification system to improve consistency, promote collaboration, and accelerate progress in preclinical research, particularly in laboratories without access to advanced sequencing technologies.

RT-qPCR was selected as the most practical method for this study due to its specificity, cost-effectiveness, and rapid analysis, making it well suited for investigating three genes with pre-designed primers in FFPE samples. Although RNA-seq offers broader exploratory capabilities, its higher costs, complexity, and extensive data analysis requirements make it less compatible with the objectives of our study, which focus on SCLC subclassification.

To mitigate the inherent limitations of RT-qPCR and ensure reliable results, we implemented careful primer optimization, stringent RNA quality control, and appropriate normalization controls. Additionally, validation using independent human samples is essential to confirm the robustness and generalizability of this method and its findings across broader contexts.

Our results also highlight the critical need for advancing clinical validation for SCLC subtype classification, which has demonstrated promising impacts in preclinical and retrospective studies to date. Further research is needed to validate RT-qPCR single-gene classification in human samples and clinically diverse SCLC cohorts, as well as to explore its potential integration into routine clinical workflows.

## 4. Materials and Methods

### 4.1. Cell Lines and Cell Cultures

Human SCLC cell lines, including H69 (ATCC number: HTB-119, RRID: CVCL_1579), Shp77 (ATCC number: CRL-2195, RRID: CVCL_1693), H748 (ATCC number: CRL-5841, RRID: CVCL_1588), H345 (ATCC number: HTB-180, RRID: CVCL_1558), H82 (ATCC number: HTB-175, RRID: CVCL_1591), H1048 (ATCC number: CRL-5853, RRID: CVCL_1453), H526 (ATCC number: HTB-119, RRID: CVCL_1569), and H196 (ATCC number: HTB-119, RRID: CVCL_1509), were obtained from the American Type Culture Collection (ATCC), while the DMS273 (ECACC number: 95,062,830, RRID: CVCL_1176), cell line was obtained from the European Collection of Authenticated Cell Cultures (ECACC). The H841 cell line (ATCC number: CRL-5845, RRID: CVCL_1595) was generously provided by Dr. Montse Sánchez (Institut Josep Carreras, Badalona, Spain). The H69M cell line was generated in our laboratory by prolonged exposure of H69 cells to exogenous HGF [34]. KP1 [24], 5B, RPP631 [25], RP-48 and RP-116 [26] cells were derived from mouse SCLC tumors that developed in the lungs of two types of GEMMs: *Trp53*^Δ/Δ^/*Rb1*^Δ/Δ^ mice, which included the KP1 and 5B cell lines (B6129SF1/J strain) and the RP-48 and RP-116 cell lines (C57BL/6J) and *Trp53*^Δ/Δ^/*Rb1*^Δ/Δ^/*Rbl2*^Δ/Δ^ mice, which included the RPP631 cell line (C57BL/6J strain). KP1 cells were kindly provided by Dr. Julien Sage (Stanford University, Stanford, CA, USA), 5B cells by Dr. Kwon-Sik Park (University of Virginia, Charlottesville, VA, USA), RPP631 by Dr. Israel Cañadas (Fox Chase Cancer Center, Philadelphia, PA, USA), and RP-48 and RP-116 by Dr. Kate Sutherland (Walter and Eliza Hall Institute of Medical Research, Parkville, Melbourne, Australia). 5B, RPP631, RP-48, and RP-116 cells were shared under a Material Transfer Agreement.

H69, Shp77, H748, DMS273, H69M, KP1, and 5B were maintained in RPMI 1640 medium supplemented with 10% FBS (Fetal Bovine Serum). H82, H526, and H196 were cultured in ATCC-modified RPMI 1640 with 10% FBS. RP-48 and RP-116 were maintained in Dulbecco’s Modified Eagle’s Medium (DMEM) supplemented with 15% FBS, 100 μM β-mercaptoethanol and 10 μg/mL insulin. H345, H1048, and H841 were maintained in DMEM:F12 supplemented with HITES and 10% FBS while RPP631 was cultured in RPMI 1640 medium with HITES and 10% FBS. The HITES supplement contained 0.005 mg/mL insulin, 0.01 mg/mL transferrin, 30 nM sodium selenite, 10 nM hydrocortisone, and 10 nM beta-estradiol. All media were further supplemented with 2 mM L-Glutamine, 1 mM sodium pyruvate, 100 units/mL penicillin, and 100 µg/mL streptomycin. Cell lines were cultured in a humidified atmosphere of 95% air and 5% CO_2_ at 37 °C.

### 4.2. Human Samples

Blood (*n* = 34) and tumor (*n* = 14) samples from patients with SCLC were obtained at Hospital del Mar (Barcelona, Spain) for the purpose of generating preclinical models. This study was approved by the Institutional Review Board (approval number 2019/8586/I; 2022/10713/I).

### 4.3. Western Blot Analysis

#### 4.3.1. Total Protein Extraction and Quantification

For whole-cell protein extracts, cells were cultured in 60 or 100 mm dishes at the optimal density for each cell line. The lysis buffer consisted of 50 mM Tris-HCl (pH = 7.4), 150 mM NaCl, 1% NP40, 5 mM EDTA, and 5 mM NaF, supplemented with 2 mM Na_3_VO_4_ and protease inhibitors (Complete Mini-EDTA-free Protease Inhibitor Cocktail, Sigma, St. Louis, MO, USA). The protocol at this stage differed between adherent cells and suspension cells. For adherent cells, the medium was removed, and cells were washed with PBS and lysed mechanically in cold lysis buffer using a scraper.

For suspension cells, they were harvested and centrifuged at 300× *g* for 3 min (minutes). After centrifugation, the medium was discarded, and the cell pellet was washed with PBS. The cells were centrifugated again at 300× *g* for 3 min, and the resulting pellet was resuspended in cold lysis buffer. At this point, both adherent and suspension cells were processed using the same protocol. Cell lysates were shaken for 20 min at 4 °C, followed by centrifugation at 13,200 rpm for 10 min at 4 °C. The supernatant was collected and stored at −20 °C.

The concentration of solubilized proteins in whole-cell lysates was determined using the colorimetric DC Protein Assay (Bio-Rad, Hercules, CA, USA), based on the Lowry method. An acidic dye was added to each sample dilution, and after a 15 min incubation at room temperature (RT) protected from light, absorbance was measured at 750 nm using a plate spectrophotometer. The protein concentration of each sample was calculated by comparing its absorbance to a standard curve generated with BSA.

#### 4.3.2. Western Blot Protein Detection

To analyze the expression of specific proteins across different cell types or experimental conditions, equal amounts of protein for cell lysates (20–25 µg) were normalized to the same final volume using lysis buffer. Each sample was then mixed with an equal volume of 2× Laemmli-sample buffer containing 5% β-mercaptoethanol and denatured at 95 °C for 5 min. The prepared samples were loaded into wells of a 7–20% SDS-PAGE (sodium dodecyl sulfate-polyacrylamide) pre-cast gel (Bio-Rad) alongside the Precision Plus Protein Dual Color Standards molecular weight marker (Bio-Rad) for electrophoresis. Subsequently, proteins were transferred onto a polyvinylidene difluoride (PDVF) (membrane (Bio-Rad) using the Trans-Blot Turbo Transfer System and Packs (Bio-Rad), which included optimized buffer, membrane, and filter paper. To ensure specific antibody binding and prevent nonspecific interactions, membranes were blocked with 5% non-fat milk in TBS-T (Tris-buffered saline with Tween 0.1%) for at least 1 h at RT with constant rocking. The membranes were then incubated overnight at 4 °C with primary antibodies (listed below) diluted in 5% BSA in TBS-T, again with constant rocking. The next day, the membranes were washed three times with TBS-T to remove unbound antibodies and incubated with horseradish peroxidase (HRP)-conjugated secondary antibodies (Amersham), diluted in 5% non-fat milk in TBS-T) for 1 h at RT with constant rocking. Subsequently, the membranes were washed three times with TBS-T, and target proteins were digitally visualized using the Vilber FUSION-FX7 imaging system, following membrane treatment with enhanced chemoluminescence (ECL; Cytiva). The following primary antibodies were used for protein detection: GAPDH (#5174 Cell Signaling, Danvers, MA, USA), Actin (A5316, Merck Life Science, Boston, MA, USA), *ASCL1* (556604, BD), *NEUROD1* (ab213725, Abcam) and *POU2F3* (NBP1-83966, Novus). Densitometry for the quantitative analysis of protein expression was performed with ImageJ software (v. 1.8.0_345). Band intensities of target proteins were normalized to the corresponding loading control to account for variations in protein loading.

### 4.4. RT-qPCR

Total RNA from cell lines was extracted using the RNeasy Mini Kit (74104, Qiagen, Hilden, Germany), while RNA from FFPE PDX and CDX samples was isolated using the RNeasy FFPE Kit (73504, Qiagen). RNA concentrations were determined using a NanoDrop ND-1000 spectrophotometer (NanoDrop Technologies, Wilmington, DE, USA), and RNA integrity (RIN) was determined using the Bioanalyzer System (Agilent, Santa Clara, CA, USA) at MARGenomics (Hospital del Mar Research Institute, Barcelona, Spain) and the TapeStation System (Agilent) at Genomics Core Facility (Universitat Pompeu Fabra, Barcelona, Spain). mRNA was subsequently reverse-transcribed into cDNA using the High-Capacity cDNA Reverse Transcription Kit (4368814, Thermo Fisher Scientific, Waltham, MA, USA), which efficiently synthesizes high-quality single-stranded cDNA from 0.02 to 2 μg of total RNA. Retrotranscription was routinely performed using 0.5 μg of RNA from cell lines. However, for lower-quality RNA extracted from FFPE samples, the input was increased to 1 μg of RNA to ensure optimal cDNA synthesis. cDNA reverse transcription was performed in a SimpliAmp Thermal Cycler (Applied Biosystems, Foster City, CA, USA) under the following conditions:10 min at 25 °C, 120 min at 37 °C, 5 min at 85 °C, followed by holding at 4 °C.

For cDNA amplification, samples were loaded in triplicate in 384-well plates along with forward and reverse primers (Table 2) ordered from Sigma-Aldrich and LightCycler^®^ 480 SYBR Green I Master mix (4707516001, Roche, Basel, Switzerland). Quantitative PCR was performed with primer amplification at 60 °C using the QuantStudio 12K Flex Real-Time PCR System (Applied Biosystems) at Universitat Pompeu Fabra (UPF) Genomics Core Facility.

Regarding the data analysis, we used the inverse of the ΔCt method, an unconventional but effective approach given the absence of a calibrator or control sample. While it does not represent the exponential nature of PCR raw data, it was chosen for its simplicity and alignment with the study’s objectives. Unlike the 2^−ΔΔCt^ method, optimized for fold-change comparisons across experimental groups, the ΔCt method provides a straightforward way to compare relative gene expression levels of multiple genes within the same sample. By normalizing target gene Ct values to a reference gene within the same sample, this approach ensures robustness against variations in RNA input quantity and reverse transcription efficiency. Presenting results as inverse fractions (1/ΔCt) improved data visualization and facilitated the interpretation of expression trends. However, this method does have limitations, as it does not establish a direct relationship with the absolute or relative RNA quantities.

### 4.5. Animals

For the KP1 and 5B SCLC syngeneic models, F1 progeny (males and females) of six- to twelve-week-old hybrid B6129SF1/J mice, obtained by crossing C57BL/6 females (Charles River Laboratories, Wilmington, MA, USA) with 129S1/SvlmJ males (The Jackson Laboratory, Bar Harbor, ME, USA), was used. For the RPP631, RP-48, and RP-116 SCLC syngeneic model, six-week-old C57BL/6J male mice were purchased from Charles River Laboratories. Six- to twelve-week-old male and female NOD-SCID immunodeficient mice were purchased from Janvier Labs for the generation of PDX and CDX models.

All mice were housed in the pathogen-free animal facility at the Barcelona Biomedical Research Park (PRBB, Barcelona, Spain). A 7-day acclimatization period was established prior to the initiation of any experiments. The protocols followed were designed in compliance with ethical regulations for animal experimentation in Spain (Spanish Legislation, Real Decreto 53/2013, BOE 34/11370–11421) and Europe (European Community Directive 2010/63/EU). Ethical approval was obtained from the Animal Research Ethics Committee of the PRBB and the Animal Welfare Department of Catalonia, Spain (EARA-20-0045-P1; EARA-20-0045-P2; JAM-23-0017-P1).

Tumor volumes in syngeneic models were monitored twice weekly. PDX and CDX models were routinely supervised based on their progression. Mice were euthanized using CO_2_ before tumors reached a volume of 1500 mm^3^ or if their health condition deteriorated.

### 4.6. Subcutaneos Mouse SCLC Models

Mouse SCLC cells (1 × 10^6^ KP1, 5B, and RP-48 cells; 2 × 10^6^ RP-116 cells; 8 × 10^6^ RPP631 cells) were collected, resuspended in sterile PBS, and mixed with Matrigel (354234, Corning) at a 1:1 ratio. The mixture was then injected into the right flank of B6129SF1/J mice (KP1 and 5B) or C57BL/6J mice (RPP631, RP-48 and RP-116). Randomization and blinding were not applied, as animals were not distributed into different groups. No animal exclusions were performed in these experiments.

### 4.7. Patient-Derived Xenografts (PDX) Generation

Depending on the type of tumor sample (tru-cut biopsy or tumor fragment) obtained from patients with SCLC, the protocol varied. In both cases (tru-cut biopsy or tumor fragment), the tumor sample was washed with PBS supplemented with 100 units/mL penicillin and 100 µg/mL streptomycin.

For tumor fragment (not tru-cut), the tumor sample was obtained post-surgery directly from the surgical resection piece, in a 1:10 HITES:PBS medium supplemented with 1% penicillin–streptomycin. Under sterile conditions, the sample was mechanically dissociated into fragments of approximately 3 × 3 mm and embedded in Matrigel (354234, Corning, NY, USA). A small incision was made on the back of each mouse with scissors, and the fragments coated with Matrigel were inserted subcutaneously into each flank through the incision using surgery forceps. The incision was closed with a staple, which was removed seven days later.

For tru-cut biopsies, the tissue sample was transferred to a 100 µm cell strainer positioned over a 50 mL tube and mechanically disaggregated using a syringe plunger. The cell strainer was then washed with 10 mL of PBS. The resulting cellular suspension was centrifuged at 300× *g* for 5 min, and the supernatant was discarded. The pellet was resuspended in 50 µL of cold PBS, followed by the addition of 50 µL of Matrigel (354234, Corning) at a 1:1 ratio. This mixture was injected subcutaneously into the right flank of mice.

### 4.8. Circulating Tumor-Cells (CDTs) Isolation and Generation of CTC-Derived Xenografts (CDX)

The CTC isolation protocol was adapted from Hodgkinson et al., 2014 [44]. Blood samples from patients with advanced SCLC were collected using three CellSave preservative tubes (7900005, Menarini, Florence, Italy) (~30 mL) and stored at RT for up to 24 h with constant mixing. A volume of 500 μL of RosetteSep Human Circulating Epithelial Tumor Cell Cocktail (Stem Cell Technology, Vancouver, Canada) was added to each tube and incubated for 20 min at RT with continuous mixing. Meanwhile, a 9:1 solution of Hank’s Balanced Saline Solution (HBSS) and RPMI HITES medium was prepared (without FBS). The blood was diluted 1:1 with HBSS: RPMI HITES solution. The diluted blood was then layered over 15 mL of Lymphoprep in 50 mL tubes and centrifuged at 1200× *g* for 20 min without a break. Cells at the interphase between Lymphoprep and plasma were collected using a sterile Pasteur pipette, diluted with cold HBSS: HITES to fill the 50 mL tube, and centrifuged at 250× *g* for 5 min. The supernatant was carefully removed, as the cell pellet was not visible. The pellet was subsequently resuspended in 1 mL of cold HBSS: HITES solution, transferred to a sterile Eppendorf tube, and centrifuged at 300× *g* for 5 min. The resulting cells were resuspended in 50 µL of cold HITES medium, combined with 50 µL of Matrigel, and maintained on ice until subcutaneous injection into the right flank of the mouse.

### 4.9. Formalin-Fixed Paraffin-Embedded (FFPE) Blocks of Biological Samples

FFPE blocks were prepared from cell line pellets, mouse SCLC syngeneic tumors, PDX models, and CDX models. Cell lines were cultured for two weeks before being harvested and centrifuged to form cell pellets. These pellets were mixed with Histogel (HG-4000-12, Fisher Scientific) which solidifies upon cooling. Once prepared, the cell pellets, along with mouse SCLC syngeneic tumors, PDX, and CDX tumor samples, underwent identical processing. All samples were fixed in 10% neutral-buffered formalin for 12 h, dehydrated using a graded alcohol series up to 100%, cleared with xylene, and embedded in paraffin under vacuum conditions (Tissue–Tek VIP, Sakura). The resulting paraffin blocks were then sectioned for subsequent analysis. Histopathology of the sections was evaluated using the standard hematoxylin and eosin (H&E) staining protocol.

### 4.10. Immunohistochemistry (IHC)

The protocols were adapted to the sample characteristics and resources available in the pathology laboratories, specialized for either animal studies or hospital-based settings.

FFPE cell pellets and mouse SCLC tumor sample blocks were sectioned into 5 µm tissue slides, mounted onto histological glass slides, deparaffinized in xylene, and hydrated. Heat-induced antigen retrieval was performed in a high-pH solution using the PT Link platform (Dako). For IHC, endogenous peroxidase activity and background staining were blocked using a peroxidase-blocking solution (Dako) and goat serum, respectively. The sections were then incubated overnight at 4 °C with the primary antibody synaptophysin (SYP) (PA1-1043, Invitrogen, 1:100) and subsequently with the ImmPRESS Peroxidase (HRP) Polymer Reagent for signal amplification. Finally, the sections were visualized using 3,3′-Diaminobenzidine (DAB) and counterstained with hematoxylin.

FFPE PDX and CDX tumor sample blocks were sectioned into 4 μm tissue slides, mounted onto histological glass slides, deparaffinized in xylene, and hydrated. Heat-induced antigen retrieval was performed using CC1 buffer (Tris-based buffer; Roche) or a high-pH solution (Agilent) on different platforms (Benchmark or Autostainer), depending on the antibody. Staining conditions for each antibody are detailed in Table 3. Prior to and after incubation with the primary antibody at RT, PDX and CDX sections were blocked for 30 min with albumin. Subsequently, they were treated with either the OPTIVIEW or Flex+ (Rabbit) signal amplification systems. Visualization was achieved with DAB staining, followed by hematoxylin counterstaining. Human SCLC cases positive for each marker served as positive controls.

The following antibodies were used for immunohistochemical studies: *ASCL1* (556604, RRID: AB_396479, BD Biosciences, San Jose, CA, USA), *NEUROD1* (ab213725, RRID:AB_2801303, Abcam, Cambridge, UK), and *POU2F3* (NBP1-83966, RRID:AB_11024500, Novus, Centennial, CO, USA). Samples were digitalized using the Philips Intellisite Pathology Solution Ultra Fast Scanner for evaluation. The entire tumor area in the sections was quantified using the H-score. Tumor cell positivity levels were categorized as low, medium, or high. The final H-score was calculated using the following formula: (% low positive) + 2 (% medium positive) + 3 (% high positive).

### 4.11. Statistical Analysis

Analyses and plots were performed using GraphPad Prism 8. Results are presented as the mean ± standard deviation (SD) or the standard error of the mean (SEM). No statistical tests were conducted to compare different groups.

## 5. Conclusions

The study highlights that while reclassification of SCLC may evolve with the discovery of new molecular markers, transcription factors such as *ASCL1*, *NEUROD1*, and POU2F3 remain essential and stable markers for SCLC subtyping. RT-qPCR provides a practical and reliable method for measuring these markers, offering a simplified and cost-effective alternative to techniques like IHC and RNA sequencing. This approach is particularly valuable for classifying experimental models, enabling researchers to consistently and efficiently subtype SCLC samples, thereby supporting robust and comparable findings across diverse studies and resource settings.

## Figures and Tables

**Figure 1 ijms-26-01293-f001:**
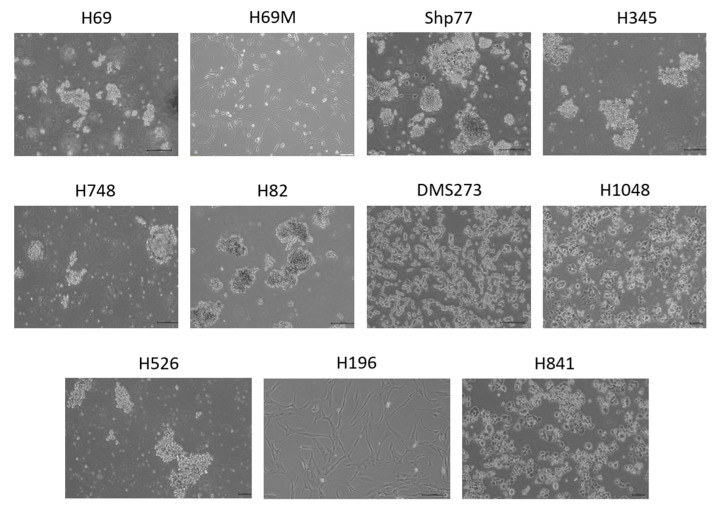
Cell morphology of human SCLC cell lines. Representative light microscopy images showing the growth patterns and cell morphology of human SCLC cells. Scale bars: 20 μm, except for H69M (100 μm).

**Figure 2 ijms-26-01293-f002:**
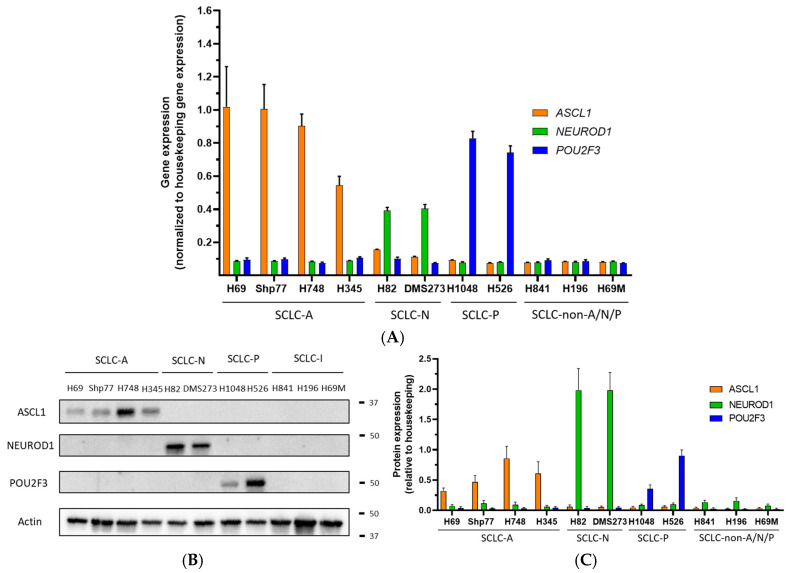
Subtype classification of human SCLC cell lines. (**A**) Basal mRNA expression levels of *ASCL1*, *NEUROD1,* and *POU2F3* in human SCLC cell lines were measured as 1/ΔCt (Ct gene−Ct *GAPDH*), with *GAPDH* as the housekeeping gene using RT-qPCR. Cells were plated and cultured for 48 h prior to RNA extraction. The plots represent the mean ± SD of three biological replicates for each cell line. (**B**) Basal protein expression analysis of *ASCL1*, *NEUROD1*, and *POU2F3* by Western blot. Cells were plated and cultured for 48 h prior to whole protein lysate extraction. Actin was used as the housekeeping control. The images are representative of three independent biological replicates. Expected molecular weights: *ASCL1* (34 kDa), *NEUROD1* (40 kDa), *POU2F3* (47 kDa), and GAPDH (37 kDa). The numbers and bands on the right indicate the protein ladder and the corresponding molecular weight. (**C**) Bar chart representing the quantitative densitometry analysis of basal protein expression levels of *ASCL1*, *NEUROD1*, and *POU2F3* by Western blot. Data are presented as mean ± SD relative protein expression to housekeeping (three independent biological replicates for each cell line).

**Figure 3 ijms-26-01293-f003:**
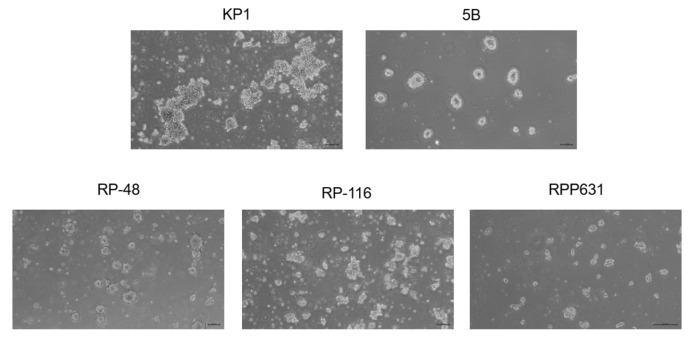
Cell morphology of mouse SCLC cell lines. Representative light microscopy images showing the growth patterns and cell morphology of mouse SCLC cells. Scale bars: 20 μm.

**Figure 4 ijms-26-01293-f004:**
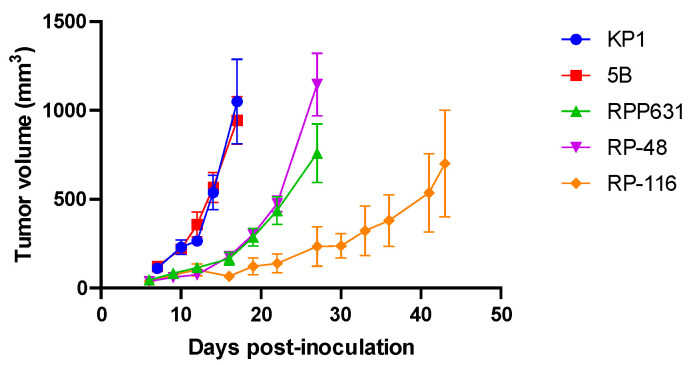
Tumor growth of subcutaneous mouse SCLC syngeneic models. Tumor growth curves were generated by plotting the mean ± SEM of tumor volumes (mm^3^) for the KP1 (*n* = 6), 5B (*n* = 5), RPP631 (*n* = 4), RP-48 (*n* = 3), and RP-116 (*n* = 4) models.

**Figure 5 ijms-26-01293-f005:**
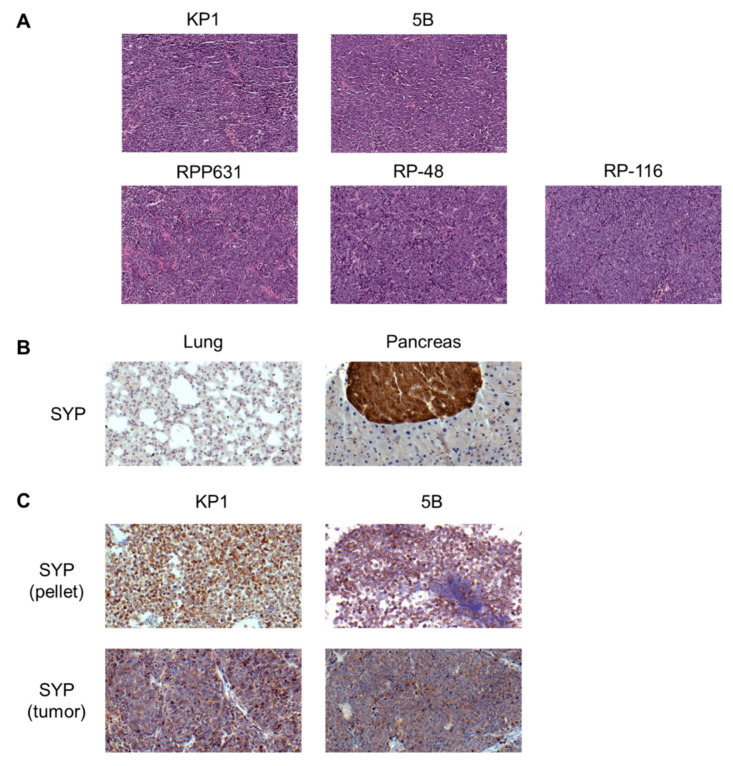
Histopathological analysis of mouse KP1, 5B, RPP631, RP-48, and RP-116 tumors. (**A**) Representative H&E images illustrating the histological morphology of the different mouse SCLC syngeneic tumors. Scale bar = 50 µm. (**B**) Representative images of negative (lung) and positive (pancreas) controls for synaptophysin staining by IHC. Scale bar = 20 µm (lung) 50 µm (pancreas). (**C**) Representative IHC images showing synaptophysin staining in cell line pellets and tumors for each cell line. Scale bar = 50 µm except KP1 pellet (20 µm). SYP: synaptophysin.

**Figure 6 ijms-26-01293-f006:**
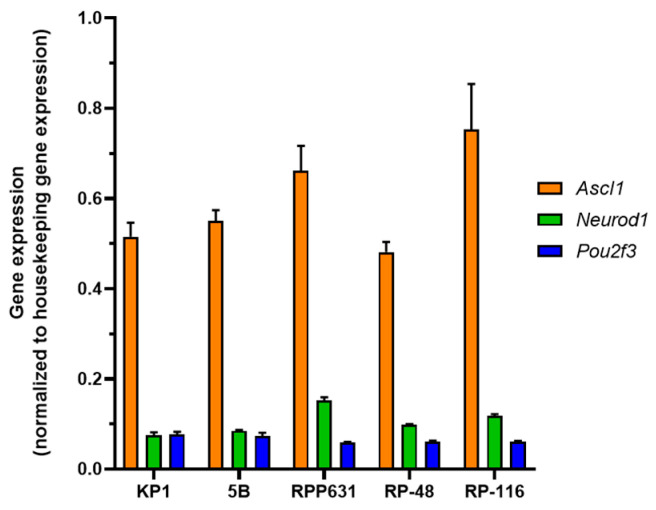
Subtype classification of mouse SCLC cell lines by RT-qPCR. Basal mRNA expression levels of *Ascl1*, *Neurod1*, and *Pou2f3* in mouse SCLC cell lines were measured as 1/ΔCt (Ct gene−Ct *Gapdh*), with *Gapdh* as the housekeeping gene. Cells were plated and cultured for 48 h prior to RNA extraction. The plots represent the mean ± SD of three biological replicates for each cell line.

**Figure 7 ijms-26-01293-f007:**
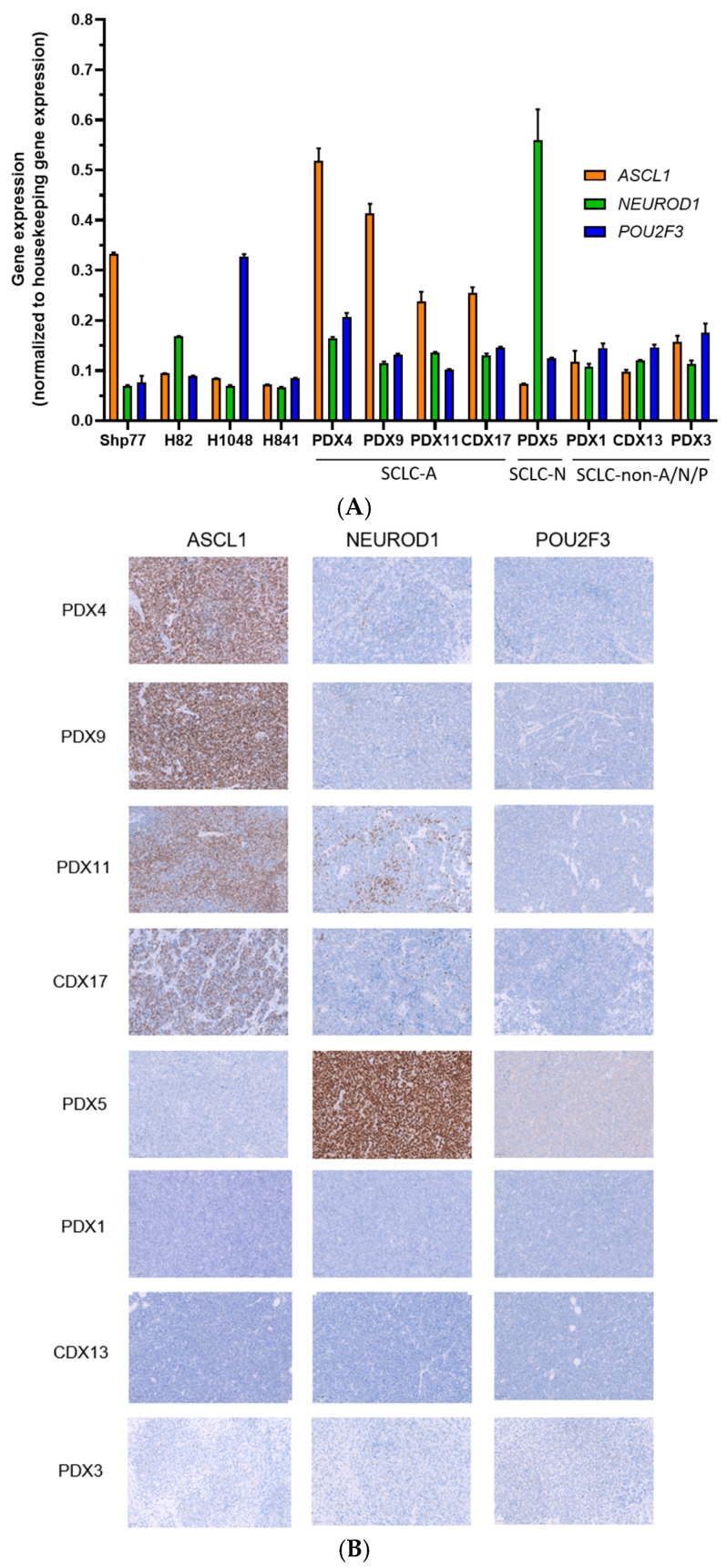
Subtype classification of PDX and CDX SCLC models by RT-qPCR and IHC. (**A**) Basal relative mRNA expression levels of *ASCL1*, *NEUROD1,* and *POU2F3* in PDX and CDX models were assessed using RT-qPCR. Gene expression is shown as 1/ΔCt (Ct gene−Ct Housekeeping), with *TOP1* used as the housekeeping gene for PDX1 and CDX13, and *TUBB* for the remaining samples. Cell lines Shp77, H82, and H1048 served as positive controls for *ASCL1*, *NEUROD1,* and *POU2F3*, respectively, while H841 was used as a control for triple-negative samples. Data are shown as the mean ± SD of three technical replicates for each sample. (**B**) Representative IHC images show staining for *ASCL1*, *NEUROD1*, and *POU2F3* in SCLC PDX and CDX models. The most prevalent marker was used to define the subtype. PDX-1, CDX-13, and PDX-3 were negative for all three markers. Staining was visualized using 3,3′-Diaminobenzidine (DAB) as the chromogen. Scale bar = 50 µm.

**Table 1 ijms-26-01293-t001:** H-score values quantified from IHC staining performed on PDX and CDX models.

	*ASCL1*	*NEUROD1*	*POU2F3*
PDX4	210	6	0
PDX9	270	6	3
PDX11	200	45	0
CDX17	120	15	0
PDX5	0	300	2
PDX1	0	0	0
CDX13	0	0	0
PDX3	9	0	3

**Table 2 ijms-26-01293-t002:** Primer sequences used for RT-qPCR of the indicated genes in human and mouse samples. Fw: forward primer; Rv: reverse primer. * used for FFPE samples.

Species	Gene	Primer Sequence (5′ → 3′)		Amplicon Size (BP)	Source
Human	*ASCL1*	Fw	TCCCCCAACTACTCCAACGAC	233	[35]
		Rv	CCCTCCCAACGCCACTG		
	*NEUROD1*	Fw	TCTTCCACGTTAAGCCTCCG	97	[36]
		Rv	CCATCAAAGGAAGGGCTGGT		
	*POU2F3*	Fw	CAGACCACCATCTCACGAT	202	[36]
		Rv	GGATGTTGGTCTCGATGC		
	*ASCL1* (short amplicon) *	Fw	CCCAAGCAAGTCAAGCGACA	77	[37]
		Rv	AAGCCGCTGAAGTTGAGCC		
	*POU2F3* (short amplicon) *	Fw	CTGCTGGAGAAGTGGCTGA	89	Designed in-house
		Rv	ACTTCACTGAGGCTGGGGTA		
	*GAPDH*	Fw	GGAGTCAACGGATTTGGTCGTA	78	[38]
		Rv	GGCAACAATATCCACTTTACCAGAGT		
	*TUBB*	Fw	CGCAGAAGAGGAGGAGGATT	116	[39]
		Rv	GAGGAAAGGGGCAGTTGAGT		
	*TOP1*	Fw	GATGAACCTGAAGATGATGGC	86	[40]
		Rv	TCAGCATCATCCTCATCTCG		
Mouse	*Ascl1*	Fw	GCTCTCCTGGGAATGGACT	70	[41]
		Rv	CGTTGGCGAGAAACACTAAAG		
	*Neurod1*	Fw	CGCAGAAGGCAAGGTGTC	90	[42]
		Rv	TTTGGTCATGTTTCCACTTCC		
	*Pou2f3*	Fw	AGCTCCTACCCCACTCTCAG	158	Designed in-house
		Rv	TCCATCGATAACTGCTCCGC		
	*Gapdh*	Fw	AGGTCGGTGTGAACGGATTTG	123	[43]
		Rv	TGTAGACCATGTAGTTGAGGTCA		

**Table 3 ijms-26-01293-t003:** Information on the antibodies used for immunohistochemistry in human samples. DAB: 3,3′-Diaminobenzidine.

Antibody	Signal Amplification	Demask	Dilution	Time (min)	Reveal
*ASCL1*	OPTIVIEW	CC1	1/200	20	DAB
*NEUROD1*	OPTIVIEW	CC1	1/100	16	DAB
*POU2F3*	Flex+	High pH	1/200	30	DAB

## Data Availability

The original contributions presented in the study are included in the article. Further inquiries can be directed to the corresponding authors.

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
