# Peer review of "The Potential of Single-Transcription Factor Gene Expression by RT-qPCR for Subtyping Small Cell Lung Cancer"

_ijms, 2025, doi:10.3390/ijms26031293_

Round 1

Reviewer 1 Report

Comments and Suggestions for Authors

ijms-3409932

The potential of single-transcription factors gene expression by RT-qPCR for subtyping Small Cell Lung Cancer

·     In this paper, the authors tested single-gene analysis by RT-qPCR from FFPE-extracted RNA aiming to simplify SCLC subtype classification and providing a cost-effective alternative to IHC staining or expensive multi-gene RNA sequencing panels, making SCLC subtyping more accessible for both preclinical research and clinical applications. Despite the overall concept is very interesting, this study contains some issues that need to be addressed:

Results

- You have used the 1 /delta ct method in determining the gene expression?  Explain the aim of using it.

- Why did you write on the y axis in figure 6 that it is normalized to GAPDH and then in figure 7 it is normalized to a house keeping gene.

- Previously it was reported that GAPDH is over expressed in this type of cancer. doi: 10.1186/1476-4598-12-97. PMID: 23988223; PMCID: PMC3766010

Have you tried to use any other housekeeping genes?

- The western blot in figure 2 B is important to show the protein expression and confirm the RTPCR results. It would be better to add a bar chart of the densitometry measurements to be able to compare between these levels.

Author Response

- You have used the 1 /delta ct method in determining the gene expression?  Explain the aim of using it.

This study analyzed the expression of three transcription factors per sample, with some expressing one factor and others none. Using RT-PCR, we assessed each target gene alongside a reference/housekeeping gene under basal conditions, without exposure to experimental variables affecting gene expression. To represent the data, we used the inverse of the ΔCt method, an unconventional yet effective approach given the absence of a calibrator or control sample. While it does not represent the exponential nature of PCR raw data, it was chosen for its simplicity and alignment with the study's objectives. Unlike the 2^(-ΔΔCt) method, optimized for fold-change comparisons across experimental groups, the 1/ΔCt method provides a straightforward way to compare relative expression levels of multiple genes within individual samples. By normalizing target gene Ct values to a reference gene within the same sample, this approach ensures robustness against variations in RNA input quantity and reverse transcription efficiency. As detailed below in response to another suggestion, the uniform expression of the reference genes was rigorously assessed, confirming their suitability as reliable housekeeping genes and ensuring methodological reliability.

Presenting results as inverse fractions (1/ΔCt) improved data visualization and interpretation of expression trends. However, this method does not directly reflect absolute or relative RNA quantities.

Future validation with independent human samples will address potential limitations in our quantification strategies by implementing alternative and comparing alternative methods for processing raw data. This remains a significant challenge, as approaches and software continue to evolve (PMID: 18812161, doi: 10.1016/j.ab.2008.08.031; PMID: 17291332, doi: 10.1186/gb-2007-8-2-r19). Nonetheless, our approach is validated by the alignment of RT-PCR results with protein-level confirmation via Western blot analysis and IHC. We thank the reviewer for the valuable suggestion regarding the ΔCt method.

In response, we have incorporated most of our explanation into the main text (Section 4.4, lines 485-497). We also clarified in the Materials and Methods section that the analysis was performed using a QuantStudio 12K Flex Real-Time PCR System (Applied Biosystems) and added details about the core facility where the RT-PCR determinations were conducted (Section 4.4., line 482).

- Why did you write on the y axis in figure 6 that it is normalized to GAPDH and then in figure 7 it is normalized to a house keeping gene.

We apologize for the mistake in figure labeling. Figure 6 has been updated to read “normalized to housekeeping gene expression,” and Figure 2 has been revised for consistency. The housekeeping gene used is now specified in each figure legend. We thank the reviewer for bringing this to our attention.

- Previously it was reported that GAPDH is over expressed in this type of cancer. doi: 10.1186/1476-4598-12-97. PMID: 23988223; PMCID: PMC3766010

Have you tried to use any other housekeeping genes?

We sincerely appreciate your reference to the study on GAPDH overexpression in NSCLC (doi: 10.1186/1476-4598-12-97). While informative, its findings may not directly apply to SCLC, which represents a distinct biological entity. A brief PubMed search did not identify comparable studies on GAPDH overexpression in SCLC.

In the absence of a universal standard for housekeeping genes, we employed the widely recommended strategy of normalizing gene expression to multiple reference genes. Following the screening process detailed below, GAPDH was selected for this study due to its stable expression and extensive use in comparable experimental frameworks. In response to your comment, we also revisited archived SCLC datasets from previous experiments which showed Tubulin amplification at 16–18 cycle thresholds (Ct) and GAPDH at 17–19 Cts, with Tubulin generally showing slightly higher expression. These minor differences did not affect result consistency due to minimal variability between the genes.

To validate our normalization approach, we analyzed the expression of β-actin, Tubulin, ATP5E, and GAPDH expression in the Shp77 cell line. The results showed highly comparable expression levels for β-actin (15.4 Cts), ATP5E (16.4 Cts), and Tubulin (16.3 Cts), with GAPDH showing only slightly lower expression (17.8 Cts). These averages, derived from three technical replicates, highlight the minimal differences between the genes and strongly support the reliability of our normalization strategy. Taking into consideration the limited sample size of our, implementing this methodology in clinical settings will require further and final optimization of the selected housekeeping gene for data analysis.

While we appreciate your recommendation, we are confident that the distinctive characteristics of our SCLC models, combined with the thorough validation performed, strongly support the suitability and robustness of the selected methodology in the context of this study.

- The western blot in figure 2 B is important to show the protein expression and confirm the RTPCR results. It would be better to add a bar chart of the densitometry measurements to be able to compare between these levels.

Thank you for the suggestion. We have added a bar chart of the densitometry measurements to Figure 2B to better compare protein expression levels and support the RT-PCR results. The figure legend (lines 173-176) and Methods Section (4.3.2,lines 461-463) have been updated accordingly, specifying that: the band intensities of target proteins were normalized to the corresponding loading control to account for variations in protein loading.

Reviewer 2 Report

Comments and Suggestions for Authors Peer review ijms 3409932

The paper entitled “The Potential of single-transcription factors gene expression by TR-qPCR for subtyping small cell lung cancer” by Iñañez et al. aims to offer a new approach for Small Cell Lung Cancer evaluation which could be more cost efficient and precise at the same time, compared to IHC or RNAseq assays.

Please find my observations below, point-by-point for each chapter.

Introduction:

This section describes the current knowledge in SCLC incidence and aggressiveness. The diagnosis methods are well described and highlight the limitations. The information provided within the introduction chapter is well written and documented, being clearly linked to the aim of this study.

I have no observations here.

Results:

The results section is well structured and easy to follow the workflow of the experiments. Subtyping for human SCLC cell lines presents the morphological features of the cells, using conventional bright field microscopy, then the molecular classification presents the cell lines that exhibited patterns of ASCL1 NEUROD1 and POU2F3. The authors clearly identified four classes, that are clustered based on the expression of these three markers. The RT-qPCR data is confirmed by the WB assays, and in this case the supplemental files are truly helpful for checking the accuracy of the data.

In terms of murine cell lines, the authors checked the tumorigenicity of these cell lines. Then the authors checked the SCLC subtype by RT-qPCR and classified the cells as SCLC-A subtype.

The RT-qPCR analysis of PDX and CDX highlighted the clustering in specific subtypes of SCLCs, which demonstrates that RT-qPCR is a feasible alternative for IHC. Of course, these findings need to be confirmed in a large cohort, to identify potential limitations or false results.

Minor comments:

Line 258 – Check if Table 3 is correct, did you meant to write table 1?

Discussion

The discussion chapter is well structured and I find it complete.

However, did the authors consider some limitations of the RT-qPCR method? Could these potential limitations be marked in the discussion section, together with the need for a validation of different cohorts?

Methods:

The materials and methods section contains almost all the details needed for other groups to reproduce the experiments; thus, I have just minor comments:

4.1. Could you please add the ATCC/ECACC number for the cell lines that you used and which were purchased from ATCC or ECACC?

4.4. What was the Yield of RNA that was used for cDNA synthesis? Is there a minimum quantity recommended for the product that you used?

For the primer synthesis, which provider did you choose? Or did you designed the primers and synthetized within your group?

Conclusions:

The conclusions are fine and in compliance with the presented results.

Overall remarks:

The paper presents a potential new diagnosis tool for SCLC and if validated on this specific pathology it may innovate the diagnosis and management for SCLC patients.
I would like to congratulate the authors for their effort in realizing this study, and I encourage them to translate this study to a patient cohort and validate their findings

Author Response

Line 258 – Check if Table 3 is correct, did you meant to write table 1?

Thank you for pointing out this error. You are correct; line 258 should reference "Table 1" instead of "Table 3." We have made the necessary corrections.

However, did the authors consider some limitations of the RT-qPCR method? Could these potential limitations be marked in the discussion section, together with the need for a validation of different cohorts

RT-PCR was selected as the most practical method for this study due to its specificity, cost-effectiveness, and rapid analysis, making it well-suited for investigating three genes with pre-designed primers in FFPE samples. Although RNA-seq offers broader exploratory capabilities, its higher costs, complexity, and extensive data analysis requirements make it less compatible with the objectives of our study, which focus on SCLC subclassification.

To mitigate the inherent limitations of RT-PCR and ensure reliable results, we implemented careful primer optimization, stringent RNA quality control, and appropriate normalization controls. Additionally, validation using independent human samples is essential to confirm the robustness and generalizability of this method and its findings across broader contexts.

We appreciate the reviewer’s suggestion, which has improved our discussion (Section 3., lines 362-372).

4.1. Could you please add the ATCC/ECACC number for the cell lines that you used and which were purchased from ATCC or ECACC?

We have added the ATCC/ECACC number for each cell line and specified their source. Additionally, we have included the Research Resource Identifier (RRID) for each cell line (Section 4.1., lines 380-389).

4.4. What was the Yield of RNA that was used for cDNA synthesis? Is there a minimum quantity recommended for the product that you used?

RNA yield varies by sample type. From 1–2 × 10^6 cells in established cell lines, we obtained 5–50 μg of RNA of high quality (RNA integrity (RIN) =10). FFPE samples typically yielded 4-6 μg of RNA from 4–5 sections (14 µm each), sufficient for analysis using our specialized protocols. Nevertheless, this type of samples presents challenges due to RNA degradation caused by formalin fixation, paraffin embedding, and suboptimal storage conditions, leading to fragmented and lower-quality RNA (even RIN <2).

For retrotranscription and cDNA synthesis, we used the Applied Biosystems™ High-Capacity cDNA Reverse Transcription Kit (catalog number: 10400745), which efficiently synthesizes high-quality single-stranded cDNA from 0.02–2 μg of total RNA. Retrotranscription was performed using 0.5 μg of RNA from cell lines. However, for lower-quality RNA extracted from FFPE samples, the input was increased to 1 μg to ensure optimal cDNA synthesis.

We appreciate the reviewer’s suggestion and have incorporated the RNA yield details described in the first paragraph of our response into the Results section (Section 2.3., lines 250-256) and the second paragraph on the Materials and Methods section (Section 4.4., lines 472-476).

For the primer synthesis, which provider did you choose? Or did you designed the primers and synthetized within your group?

Most primer sequences were obtained from published articles cited in the updated manuscript, and all were ordered from Sigma-Aldrich, as noted now in the Materials and Methods Section (Section 4.4., line 480). References for housekeeping genes used for normalization have also been included. For POU2F3, primers for human samples were available; however, primers for murine samples and human short amplicons were designed by us and validated using cell line samples identified as this subtype with the standard primers. A new “Source“ column has been added to Table 2 to clarify the origin of the sequence used.

Round 2

Reviewer 1 Report

Comments and Suggestions for Authors

The response to the questions and the amendments made were sufficient to be published.